# Haplotype in *SERPINA1* (AAT) Is Associated with Reduced Risk for COPD in a Mexican Mestizo Population

**DOI:** 10.3390/ijms21010195

**Published:** 2019-12-27

**Authors:** Marco Antonio Ponce-Gallegos, Gloria Pérez-Rubio, Adriana García-Carmona, Jesús García-Gómez, Rafael Hernández-Zenteno, Alejandra Ramírez-Venegas, Ramcés Falfán-Valencia

**Affiliations:** 1HLA Laboratory, Instituto Nacional de Enfermedades Respiratorias Ismael Cosio Villegas, Mexico City 14080, Mexico; marcoapg@iner.gob.mx (M.A.P.-G.); glofos@yahoo.com.mx (G.P.-R.); adri_garcia_168@hotmail.com (A.G.-C.); jesus.garcia3232@alumnos.udg.mx (J.G.-G.); 2Tobacco Smoking and COPD Research Department, Instituto Nacional de Enfermedades Respiratorias Ismael Cosio Villegas, Mexico City 14080, Mexico; rafherzen@yahoo.com.mx

**Keywords:** COPD, tobacco smoking, biomass burning, SNPs, *SERPINA1*, PiS, PiZ

## Abstract

Protease inhibitor S (PiS) and protease inhibitor Z (PiZ) variants in the *SERPINA1* gene are the main genetics factors associated with COPD; however, investigations about other polymorphisms are scanty. The aim of this study was to evaluate two missense single nucleotide polymorphisms (SNPs) (rs709932 and rs1303) in the *SERPINA1* gene in Mexican mestizo patients with chronic obstructive pulmonary disease (COPD) related to tobacco smoking and biomass-burning exposure. 1700 subjects were genotyped and divided into four groups: COPD related to tobacco smoking (COPD-S, *n* = 297), COPD related to biomass-burning exposure (COPD-BB, *n* = 178), smokers without COPD (SWOC, *n* = 674), and biomass-burning exposed subjects (BBES, *n* = 551) by real-time PCR. Moreover, the patients’ groups were divided according to their exacerbations’ frequency. We carried out a haplotype analysis. We did not find differences in allele and genotype frequencies between groups in unadjusted and adjusted analyses, neither with these SNPs and lung function decline. Exacerbations’ frequency is not associated with these SNPs. However, we found a haplotype with major alleles (CT) associated with reduced risk for COPD (*p* < 0.05). Our analysis reveals that SNPs different from PiS and PiZ (rs709932 and rs1303) in the *SERPINA1* gene are not associated with COPD and lung function decline in a Mexican mestizo population. However, a haplotype shaped by both major alleles (CT haplotype) is associated with reduced risk for COPD.

## 1. Introduction

In accordance with the Global Initiative for Chronic Obstructive Lung Disease (GOLD), Chronic Obstructive Pulmonary Disease (COPD) is a common, preventable, and treatable disease characterized by persistent respiratory symptoms and airflow limitation. These conditions are mainly caused by significant exposure to noxious particles or gases, being the most important tobacco smoking and biomass-burning exposure [1].

Nowadays, COPD is considered the fourth leading cause of death worldwide [1]. The Global Burden of Disease Study from 2002 to 2030 reports a prevalence of 251 million cases of COPD globally in 2016. Prevalence in Mexico was estimated at 7.8% by PLATINO study in 2005 [2,3].

Risk factors to develop COPD have been widely described. Alpha-1 antitrypsin deficiency (AATd) is the most important genetic risk factor associated with COPD [4]. AAT is a 52 kDa sialoglycoprotein encoded by the *SERPINA1* gene whose one of principal function is to protect the lower respiratory tract of lungs from proteolytic degradation by neutrophil elastase [5]. The common genotype for *SERPINA1* (AAT) is described as PiMM and by protein serum levels between 150 and 350 mg/100 mL. PiS and PiZ are the genetic variants characteristic of AATd patients. It has been previously described that individuals carrying on SS, SZ, and ZZ genotypes express serum protein concentrations of 85%, 25%, and 15%, respectively [6].

Despite PiS and PiZ variants represent approximately 95% of the AATd cases, there are other rare variants codified in the *SERPINA1* gene, such as Siiyama, Mmalton, Mprocida, Mheerlen, Mmineral springs, Mnichinan, Pduarte, Wbethesda Zaugsberg, and Zbristol that could confer risk to COPD, but there are not enough studies to support it [5].

Most of the studies of AATd have been realized in subjects with COPD related to tobacco smoking. There are no studies in patients with COPD related to biomass-burning exposure. In Mexican mestizo population, Pérez-Rubio et al. [7] studied two single nucleotide polymorphisms (SNPs) in *SERPINA1* (rs28929474 [PiZ] and rs17580 [PiS]), finding that homozygous and heterozygous (to minor allele in each case) are in a very low frequency in Mexican population. However, those subjects in the heterozygous state had poorer lung function measurements. On the other hand, Fernández-Acquier and colleagues [8] identified genetic variants different from PiS and PiZ and low serum AAT levels.

Our aim was to study two SNPs in *SERPINA1* (rs709932 and rs1303) different from PiS and PiZ in a Mexican mestizo population with COPD related to tobacco smoking and biomass-burning exposure.

## 2. Results

One thousand and seven hundred subjects were included in two cases groups and two control groups; 297 patients with COPD related to tobacco smoking (COPD-S), 178 patients with COPD related to biomass burning smoke exposure (COPD-BB), 674 smokers without COPD (SWOC), and 551 biomass-burning exposed subjects without COPD (BBES).

### 2.1. Demographic Variables

We found statistical differences in age (cases are older than controls in both comparisons), sex (in the COPD-S group subjects are predominantly men; while in COPD-BB women predominate), body mass index (BMI, COPD subjects in both comparisons have a lower BMI than their respective controls), biomass exposure index (BEI), smoking status, and pulmonary function (*p* < 0.05). Demographic and clinical data are shown in Table 1.

In addition, a comparison between COPD-S vs. COPD-BB was performed. We found that COPD-BB patients are older than COPD-S and predominantly women (*p* < 0.05). Moreover, we found that patients in COPD-S are in more advanced GOLD stages and have worse pulmonary function than the COPD-BB group (*p* < 0.05). These results are shown in Appendix A.

### 2.2. Demographic Variables in COPD Related to Tobacco Smoking and Biomass Burning and Exacerbations Frequency

We compared frequent-exacerbators (FE-S and FE-BB) against non-exacerbators (NEX-S and NEX-BB) within the COPD-S and COPD-BB groups and results are shown in Table 2. Age, sex, and smoking status did not show significant differences between FE-S and NEX-S (*p* > 0.05). However, comparing FE-BB vs. NEX-BB we found significant differences in age and BEI (*p* < 0.05). Interestingly, GOLD III and IV stages (G2) were more prevalent in the FE-S group, while GOLD I and II (G1) were more frequent in the NEX-S group. However, there were no significant differences between groups (*p* > 0.05). There were no significant differences when comparing G1 and G2 from FE-BB and NEX-BB groups, as well as there were no differences in lung function.

Furthermore, we performed a comparison between FE-S vs. NEX-BB, finding that FE-S are predominantly men, are in more advanced GOLD stages, and have worse pulmonary function than FE-BB. The results are shown in Appendix A. In addition, results of logistic regression analysis by co-variables in the COPD-S and COPD-BB groups are shown in the Appendix A, respectively.

### 2.3. Allele and Genotype Association

Two SNPs in *SERPINA1* (rs709932 and rs1303) were evaluated in this study. We did not find any association with alleles and genotypes, as well as neither with dominant nor recessive models (*p* > 0.05). Table 3 shows the allele and genotype frequencies for both comparisons. Interestingly, the minor allele frequency (MAF) of rs709932 among smokers (COPD-S and SWOC) is twice higher than reported by Ensembl for Mexican from Los Angeles, in contrast with biomass-exposed groups, which is very similar. MAF of rs1303 is a little higher in four groups than reported by Ensembl.

Due to the differences found in the clinical and demographic variables, we decided to make an analysis adjusted by covariables (age and tobacco index for smokers’ groups and age and biomass exposure index for those groups exposed to biomass burning) performing a logistic regression test.

Adjusting only by age and tobacco index in COPD-S vs. SWOC, we found statistically significant results for rs1303 (*p* = 5.82 × 10^−27^, OR = 1.12, CI 95% 1.10–1.15; *p* = 7.44× 10^−15^, OR = 1.03, CI 95% 1.03–1.04, respectively) and rs709932 (*p* = 4.16 × 10^−7^, OR = 1.12, CI 95% 1.10–1.15; *p* = 1.09 × 10^−14^, OR = 1.03, CI 95% 1.03–1.04, respectively). However, in the additive model, these associations are not maintained in both SNPs (*p* ≥ 0.05). This effect is the same for COPD-BB vs. BBES comparison adjusting by age and BEI, with significant differences with covariables alone in rs1303 (*p* = 1.25 × 10^−13^, OR = 1.08, CI 95% 1.06–1.10; *p* = 0.0003, OR = 1.002, CI 95% 1.001–1.003, respectively) and rs709932 (*p* = 2.24 × 10^−13^, OR = 1.08 CI 95% 1.06–1.10; *p* = 0.0004, OR = 1.002, CI 95% 1.001–1.003, respectively) and in additive model, these associations are not maintained in both SNPs (*p* ≥ 0.05). These results are shown in Appendix A.

### 2.4. Allele and Genotype Association with Exacerbations in COPD Related to Tobacco Smoking and Biomass-Burning Exposure Subjects

Another analysis that was carried out was a comparison between FE-S vs. NEX-S and FE-BB vs. NEX-BB. Allele and genotype frequencies did no show statistically significant differences between groups and neither did dominant and recessive models (*p* > 0.05). The results are shown in Table 4.

### 2.5. SNPs Haplotypes in SERPINA1

The haplotype analysis was conducted to determine its association with COPD susceptibility. The analysis included the two SNPs in the *SERPINA1* gene, comparing COPD-S vs. SWOC and COPD-BB vs. BBES. Polymorphisms evaluated met Hardy–Weinberg equilibrium (*p* > 0.05). Haplotypes are shown in Figure 1. Figure 1A shows the haplotype in the comparison of COPD-S vs. SWOC. This haplotype is not in high linkage disequilibrium (D’ = 89; r^2^ = 20). However, CT haplotype is associated with reduced risk for COPD (OR = 0.81, *p* = 0.048, CI 95% 0.66–0.99). Haplotype generated in COPD-BB vs. BBES comparison is shown in Figure 1B. This haplotype neither is in high linkage disequilibrium (D’ = 85; r^2^ = 9). None of the haplotypes were associated.

### 2.6. SERPINA1 Variants and Lung Function

We did not find any association with decreased lung function in COPD-S and COPD-BB groups and genotypes for both SNPs. Results are shown in Appendix A.

## 3. Discussion

In our study, we tried to identify two SNPs in the *SERPINA1* gene that could confer risk for developing COPD related to tobacco smoking and biomass-burning, as well as their association with frequent exacerbator phenotype and lung function. It has been widely described that AATd is the most important genetic risk factor to develop COPD. Reduced circulating levels of alpha-1 antitrypsin leads to increased neutrophil elastase activity in the lungs, which results in lung remodeling due to protease–antiprotease imbalance [9].

Regarding demographic variables, in the comparison between COPD-S vs. SWOC, the prevalence of the disease is higher in men, while in COPD-BB vs. BBES disease is more prevalent in women, similar to previous reports [3,10]; previously we have described [11] these differences in sex distribution between groups, which are due to historical and sociocultural backgrounds. Since it is common in rural regions to find a greater number of people exposed to BBS, and at the same time, this exposed group are women in charge of household chores (mainly cooking), activity in which usually use some kind of biomass to start and maintain the combustion to heat meals. For this reason, the biomass burning-smoke group had a predominance of females.

Moreover, in both comparisons, cases are older and have higher values for tobacco-smoking variables and biomass-burning and poor lung function than their respective controls, which is expected because it is the most important parameter to differentiate between cases and controls.

Arkhipov et al. [12] described that GOLD II and III stages are the most prevalent, coinciding with our findings in both cases groups. We did not find differences in age in FE-S vs. NEX-S and FE-BB vs. NEX-BB comparisons. Interestingly, those patients in G2 tend to present more exacerbations per year in the FE-S group. In FE-BB vs. NEX-BB, there were no significant differences. These findings are consistent with those reported by large prospective cohorts such as SUPPORT trial, SPIROMICS, and ECLIPSE, where they described that those patients in GOLD III and IV have more exacerbations per year. However, they also comment that COPD is very variable and that in a follow-up of three years only a low percentage remains as FE, and those who are classified as NEX can also change to FE [12,13,14] due to the multiple factors that participate in its appearances, such as age, disease severity, poor lung function, use of inhaled corticosteroids, and at least one exacerbation in last year [15].

In 1999, Hill et al. [16] described that patients with AATd had lower AAT sputum levels and secretory leukoprotease inhibitor (SLPI), as well as higher elastase activity at the start of exacerbation, compared with COPD patients without deficiency, suggesting that patients with AATd have a higher risk for severe exacerbations. The importance of exacerbations in COPD patients reside in the fact that they generate a decrease in quality of life, promote worsening and progression of the disease, and are the main cause of death in patients with COPD.

In our analysis of allele and genotype association with COPD, rs709932 and rs1303 did not show statistically significant results in any of both comparisons. There are few reports about these SNPs and their association with COPD. One of them, in accordance with our results, Fujimoto et al. [17] evaluated 12 SNPs in *SERPINA1* (rs8004738, rs17751769, rs709932, rs11832, rs1303, rs28929474, and rs17580), *SERPINA3* (rs4934, rs17473, and rs1800463), and *SERPINE2* (rs840088 and rs975278) and did not find association with rs709932 and rs1303, only did with *SERPINE2* SNPs.

Other studies have evaluated other genetic variants in *SERPINA1*. For example, Quint and colleagues [18] reported that *SERPINA1* 11478G→A variant is not associated with a major risk for developing COPD in a UK population. On the other hand, Deng et al. [19] described an association with a higher risk for COPD with rs8004738 (G/A) in a Chinese population. Our research group previously reported that PiZ (rs28929474, G/A) and PiS (rs17580, A/T) were not associated with COPD in a Mexican mestizo population. However, there was an association with rs17580 and decreased FEV1/FVC ratio. Interestingly, in a subsequent analysis of increasing sample size, it was described that heterozygous genotype (AT) in rs17580 is associated with a higher risk for COPD, being more important in COPD related to biomass-burning smoke (OR = 11.5) [20]. There are no previous studies that evaluate *SERPINA1* SNPs difference from PiS and PiZ and their association with COPD related to biomass-burning smoke.

Previous studies have tried to find an association with SNPs and exacerbations frequency, due to the clinical importance in patients with COPD, with positive results in chemokine ligand 1 [21] and surfactant protein B and D [22]. Conversely, Dicker et al. [23] and Platé and colleagues [24] described a reduced risk for exacerbations with SNPs in *MBL* and *PAR1,* respectively. There is only one previous study that evaluates SNPs in *SERPINA1* and its association with exacerbations frequency; similar to our results, authors did not find differences in genotypes between frequent and infrequent exacerbators and neither with AAT serum levels [18]. In addition, Ingebrigtsen et al. [25] found that having at least one plasma α1-antitrypsin-lowering Z-allele (MZ, SZ, or ZZ) was associated with increased risk for COPD exacerbations. In contrast, the S allele was not associated with exacerbations.

Interestingly, we found a haplotype associated with reduced risk for COPD related to tobacco smoking. To our knowledge, it has been never reported before. However, Chappel and colleagues [26] evaluated 10 SNPs in *SERPINA1* (including rs709932) and five in *SERPINA3* and they found six haplotypes in *SERPINA1* that confer risk of disease by six- to 50-fold. On the other hand, this is the first study that assesses rs709932 and rs1303 in COPD related to biomass-burning exposure, so we do not have a benchmark in this case.

Despite S and Z alleles have been described as the most common risk variants for COPD, some other studies have found risk variants different from PiS and PiZ. For example, Echazarreta et al. [27] measured the plasmatic levels of AAT in Argentinian subjects with chronic pulmonary diseases and pneumothorax and found AATd in almost a quarter of their study group. After genotyping, 7 patients with ZZ, 3 with SZ, 2 with SS, 8 with MZ, and 33 with MS genotype were found. However, most of the subjects evaluated with AATd showed a different genotype than S or Z. In addition, Kevorkof et al. [28] and Sorroche and colleagues [29] in their respective cohorts also found a high number of patients with genotype different than S or Z with AATd. These studies show the importance of evaluating different variants that could be involved in the development of COPD.

Previous studies have identified rs709932 and rs1303 as PiM2 and PiM3 variants, respectively [30,31,32]. In that context, despite these alleles are considered as “normal”, in a Tunisian population, PiM2 allele has been identified as a risk factor for COPD with higher frequency in COPD patients than controls [33]. Moreover, Gupta et al. [34] in an Indian population described that PiM3 was more frequent in COPD patients than healthy controls. Thus far, how this normal allele variants, which expressing normal levels of AAT, participate in the pathogenesis of COPD is unclear [30,34].

There are several studies that have tried to identify SNPs associated with lung function decline with controversial results. Genetic variants in *NLRP*, *MMP3*, *IL8*, *TIMP1,* and *EDN1* have been previously associated with the worsening of COPD and FEV1 decline in different populations [35,36,37,38,39]. In addition, Gian-Andri et al. [40] described that in the SAPALDIA cohort those subjects who carry on at least one Z allele had a higher risk for accelerate lung function decline, especially in population subgroups characterized by low-grade inflammation. In agreement with our results, Quint and colleagues [18] neither found significant differences in lung function associated with 11478G→A polymorphism. These findings suggest that rs1303 and rs709932 are not associated with lung function decline in COPD related to tobacco smoking and biomass burning.

Due to the important number of patients with lower levels of AAT without known risk-alleles, such as S and Z, it is necessary later investigations to evaluate how these variants participate in COPD pathogenesis and to offer a better diagnosis and treatment.

This study is not free of limitations. Maybe the most important is that we were not able to measure AAT levels. Furthermore, this is a retrospective study and can have memory bias (in expositional variables referred by participants); the sample size was reduced when we realized the exacerbations analysis due to the missed data in clinical records; finally, for the rs1303, the statistical power is <80% probably due its relatively low allele frequency in the studied population, which must be taken into consideration and possibly require later population validation. Among strengths are the sample size, which is large, and we have two comparison groups (COPD related to tobacco smoking and biomass-burning smoke exposure). Moreover, this is the first time that rs709932 and rs1303 are evaluated in COPD related to biomass-burning smoke.

## 4. Materials and Methods

### 4.1. Study Population

#### Case and Control Groups

One thousand and seven hundred subjects were included in this case-control study. These subjects attended the COPD and smoking cessation support clinics, both are part of the Department of Smoking and COPD Research Department of the Instituto Nacional de Enfermedades Respiratorias Ismael Cosio Villegas (INER), Mexico.

The sample included 297 patients with the diagnosis of COPD related to tobacco smoking (COPD-S) and 178 with COPD related to biomass-burning exposure (COPD-BB). The COPD diagnosis was confirmed using pulmonary function tests considering a ratio of forced expiratory volume in the first second/forced vital capacity (FEV_1_/FVC) <70% after post-bronchodilator use according to the reference values for Mexicans obtained by Perez-Padilla et al. [41]. Individuals ≥50 years, with a tobacco index ≥10 packs/year and a history of never have been exposed to biomass-burning, were classified in the COPD-S group, and those ≥50 years, with an exposure index ≥100 h/year to biomass-burning exposure, never-smokers, were classified into the COPD-BB group. In addition, the patients’ groups (COPD-S and COPD-BB) were divided into two subgroups: frequent-exacerbators (FE-S and FE-BB, respectively) (≥2 exacerbations per year) and non-exacerbators (NEX-S and NEX-BB, respectively) with emphysema or chronic bronchitis (<2 exacerbations per year), according to Spanish COPD Guidelines (GesEPOC) [42]. Exacerbations were diagnosed and classified according to Anthonisen criteria [43]. In addition, GOLD I and II stages were grouped as G1, and III and IV stages as G2. Exacerbations history was obtained from medical records and those patients who did not have information about exacerbations were excluded from this analysis. Consecutive COPD patients were enrolled from the COPD support clinic attending from 2009 to 2015.

The control groups were according to cases exposition risk factors as follows: a group of smokers without COPD (SWOC, *n* = 674), with no evidence of pulmonary disease and normal spirometry parameters. Finally, a group of biomass-burning exposed subjects (BBES, *n* = 551) without a background of active or passive smoking and no evidence of pulmonary disease and normal spirometry parameters, was also included.

Cases and controls of biomass-burning exposure are part of the national program for equality between women and men with the “Diagnóstico oportuno de EPOC/Respirar sin humo” campaign in women living in rural areas, mainly in the northern highlands of the state of Oaxaca and suburban [44] areas of the Tlalpan mayoralty of Mexico City. Subjects attending campaigns from 2014 to 2018 were included.

All participants underwent a background questionnaire of inherited pathologies, whereby subjects who reported suffering some type of lung and/or chronic inflammatory disease were excluded, as well subjects with non-Mexican ancestry (with no Mexican-birth parents and grandparents). A brief flowchart of the subjects’ selection is shown in Appendix A.

In addition, participants were previously invited to participate in the study; they signed an informed consent document and were provided with a privacy statement describing the protection of personal data. Both documents were approved by the Science and Research Bioethics Committee of this institute (approbation protocol codes: B10-12 and B11-19, approved on May 2014 and May 2019, respectively). All experiments were performed in accordance with the relevant guidelines and regulations. The STREGA (STrengthening the REporting of Genetic Association) guidelines were taken into consideration in the design of this genetic association study.

### 4.2. DNA Extraction

The DNA was extracted from peripheral blood cells via venipuncture, using the commercial BDtract Genomic DNA isolation kit (Maxim Biotech, San Francisco, CA, USA). The DNA was then quantified by UV absorption spectrophotometry at the 260-nm wavelength using a NanoDrop system (Thermo Scientific, Wilmington, DE, USA).

### 4.3. SNPs’ Selection

SNPs were selected based on bibliographic search, identifying polymorphisms previously associated with other inflammatory and respiratory diseases. Moreover, we considered an allelic frequency (MAF) higher than 5%. SNPs evaluated were rs709932 and rs1303 (*SERPINA1*). Table 5 summarizes the principal characteristics of evaluated SNPs.

### 4.4. Genotyping of the SNPs

The allelic discrimination of SNPs was performed using the commercial TaqMan probes (Applied Biosystems, San Francisco California, USA) at a concentration of 20X. We used the technique of qPCR in a 7300 Real-Time PCR System kit (Applied Biosystems, San Francisco, CA, USA), and the analysis was performed by the SDS (sequence detection software) version 1.4 software (Applied Biosystems, San Francisco, CA, USA).

Haploview version 4.2 was used to determine the presence of haplotypes in the *SERPINA1* gene associated with COPD susceptibility.

### 4.5. Lung Function

According to ambient risk factors (tobacco smoking or biomass-burning smoke exposure), patients were divided into two groups: patients with the common genotype and subjects heterozygous + homozygous to the minor allele for each SNP (rs1303: TT vs. TG vs. GG and rs709932: CC vs. CT + TT). The Median test was used to compare lung function values.

### 4.6. Statistical Analysis

The differences between groups under study were evaluated by determining and comparing the allele, genotype, and haplotype frequencies. Statistical significance was assessed using the SPSS v20.0 (SPSS Inc., Chicago, IL, USA) and Epi Info 7.1.4.0 (Centers for Disease Control and Prevention, Atlanta, GA, USA) statistical software, considering the χ^2^ values, as well as Mann–Whitney U test to comparing cases and control groups. The results were considered significant when the *p*-value was <0.05; similarly, the odds ratios (OR) with 95% confidence intervals (CI) were estimated to determine the strength of the association. A logistic regression analysis was carried out to adjust by potential confounding variables using Plink v. 1.07 [45].

## 5. Conclusions

In conclusion, SNPs in *SERPINA1* different from PiS and PiZ are not associated with COPD related to tobacco smoking and biomass-burning susceptibility neither with frequent exacerbations. However, a haplotype (CT haplotype, from rs709932 and rs1303, respectively) is associated with reduced risk for COPD related to tobacco smoking.

## Figures and Tables

**Figure 1 ijms-21-00195-f001:**
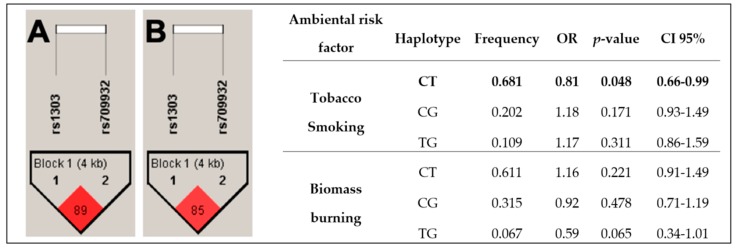
Haplotype analysis: Two SNPs were evaluated in both comparisons; among tobacco-smoking groups, the CT haplotype was identified associated with reduced risk. Panel A = COPD-S vs. SWOC; Panel B = COPD-BB vs. BBES. *p*-value < 0.05 was considered significative.

**Table 1 ijms-21-00195-t001:** Demographic variables and pulmonary function data from the four groups.

Variable	COPD-S (*n* = 297)	SWOC (*n* = 674)	*p*	COPD-BB (*n* = 178)	BBES (*n* = 551)	*p*
Age, years	67 (50–80)	55 (35–90)	<0.0001	74 (51–86)	59 (34–90)	<0.0001
Male, %	76.35	44		8	0.5	
BMI	24.5 (13.5–36.1)	27.5 (13.7–47.0)	<0.0001	24.5 (18.7–32.7)	28.2 (15.7–44.9)	0.008
**Biomass exposure status**	NA	NA				
Years exposed				50 (10–80)	40 (1–87)	<0.0001
Hours per day				7 (2–24)	5 (1–24)	<0.0001
Biomass exposure index				360 (140–828)	200 (2–1050)	<0.0001
**Smoking status**				NA	NA	
Years of smoking	42 (10–68)	34 (4–65)	<0.0001			
Cigarettes per day	20 (5–80)	14 (1–100)	<0.0001			
Tobacco index	40 (10–232)	20 (1–175)	<0.0001			
Current smokers, %	69.59	20.3	<0.0001			
**GOLD**						
GOLD I, %	14.73			27.21		
GOLD II, %	44.52			54.41		
GOLD III, %	29.11			15.44		
GOLD IV, %	11.64			2.94		
**Lung function**						
FVC (%) post	78 (24–155)	95 (18–139)	<0.0001	79 (43–144)	100 (53–193)	<0.0001
FEV1 (%) post	42 (12–88)	98 (23–142)	<0.0001	48 (18–94)	105 (53–187)	<0.0001
FEV1/FVC (%) post	52 (26.1–69.4)	103.7 (70–132.8)	<0.0001	60.9 (36.7–68.4)	105 (70–142.4)	<0.0001

COPD-S = Patients with COPD related to smoking, SWOC = Smokers without COPD, COPD-BB = Patients with COPD related to biomass burning, BBES = Biomass burning-exposed subjects, NA = Not apply. All values are expressed as median. Measures of lung function are post-bronchodilator use. *p*-value < 0.05 was significative. We used the median test to make comparisons between groups. Values are expressed as minimum and maximum.

**Table 2 ijms-21-00195-t002:** Demographic variables among chronic obstructive pulmonary disease (COPD) patients FE-S, NEX-S, FE-BB, and NEX-BB.

Variable	FE-S (*n* = 32)	NEX-S (*n* = 87)	*p*	FE-BB (*n* = 38)	NEX-BB (*n* = 47)	*p*
Age, years	67 (52–77)	65 (47–80)	0.855	68 (61–86)	75 (51–86)	0.014
Male, %	78.12	75.86	0.796	10.53	6.38	0.489
Body mass index	24.6 (15.2–35.9)	24.8 (16.9–36.1)	0.88	23.8 (20–30.3)	24.9 (18.7–31.1)	0.824
**Biomass burnings status**	NA	NA				
Years exposed				45 (10–75)	50 (14–80)	0.406
Hours per day				6 (2–18)	8 (2–24)	0.820
Biomass Exposure Index				300 (140–800)	375 (144–828)	0.025
**Smoking status**				NA	NA	
Years of smoking	40 (30–65)	40 (15–64)	0.834			
Cigarettes per day	20 (10–40)	20 (5–40)	0.859			
Packs-year history	39.5 (16–100)	40 (10–106)	0.7			
**GOLD**						
G1 (I–II), %	43.75	63.22	0.056	81.48	79.41	0.839
G2 (III–IV), %	56.25	36.78		18.52	20.59	
**Lung function**						
FVC (%) post	76.5 (32–121)	84 (37–155)	0.172	87 (69–135)	76 (49–127)	0.469
FEV1 (%) post	35.5 (14–66)	44 (18–83)	0.164	51 (34–60)	47 (18–81)	0.891
FEV1/FVC (%) post	48.4 (29.1–65.8)	52.4 (26.1–69.4)	0.45	55.6 (37–69)	63 (36.7–69.4)	0.44

FE-S = Patients with COPD related to tobacco smoking frequent exacerbators, NEX-S = Patients with COPD related to tobacco smoking non-exacerbators, FE-BB = Patients with COPD related to biomass burning frequent exacerbators, NEX-BB = Patients with COPD related to biomass burning non-exacerbators. G1 = GOLD I and GOLD II stages, G2 = GOLD III and GOLD IV stages. NA = Not apply. All values are shown as median. Measures of lung function are post-bronchodilator use. *p*-value < 0.05 was significative. We used the median test and Fisher exact test. Values are expressed as minimum and maximum.

**Table 3 ijms-21-00195-t003:** Allele and genotype frequencies of *SERPINA1* single nucleotide polymorphisms (SNPs) evaluated.

Genotype/Allele	COPD-S		SWOC		COPD-BB		BBES	
	*n* = 297	GF/AF (%)	*n* = 674	GF/AF (%)	*n*= 178	GF/AF (%)	*n*= 551	GF/AF (%)
rs709932								
CC	224	75.42	537	79.67	156	87.64	471	85.48
CT	69	23.23	124	18.40	22	12.36	74	13.43
TT	4	1.35	13	1.93	0	0	6	1
C	517	87.04	1198	88.87	334	93.82	1016	92.20
T	77	12.96	150	11.13	22	6.18	86	7.80
CC	224	75.42	537	79.79	156	87.64	471	85.48
CT+TT	73	24.58	137	20.36	22	12.36	80	14.68
CC+CT	293	98.65	661	98.07	178	100	545	98.91
TT	4	1.35	13	1.93	0	0	6	1.09
rs1303								
TT	131	44.11	338	50.15	80	44.94	209	37.93
GT	130	43.77	271	40.21	73	41.01	250	45.37
GG	36	12.12	65	9.64	25	14	92	16.70
T	392	65.99	947	70.25	233	65.45	668	60.62
G	202	34.01	401	29.75	123	34.55	434	39.38
TT	131	44.11	338	50.15	80	44.94	209	37.93
GT+GG	166	55.89	336	49.85	98	64.05	342	74.51
TT+GT	261	87.88	609	91.17	153	85.96	459	83.30
GG	36	12.59	65	9.73	25	14.04	92	16.70

COPD-S = Patients with COPD related to tobacco smoking, SWOC = Smokers without COPD, COPD-BB = Patients with COPD related to biomass-burning exposure, BBES = Biomass-burning exposed subjects. GF = Genotype frequency, AF = Allele frequency.

**Table 4 ijms-21-00195-t004:** Analysis of the association of alleles and genotypes in codominant, dominant, and recessive models between Frequent exacerbators and Non-exacerbators.

SNP/Model	Genotype/Allele	FE-S		NEX-S		*p*-Value	OR	95% CI	FE-BB		NEX-BBES		*p*-Value	OR	95% CI
		*n* = 32	GF/AF (%)	*n* = 87	GF/AF (%)				*n* = 38	GF/AF (%)	*n* = 47	GF/AF (%)			
rs709932															
Codominant	CC	25	78.13	63	72.41	0.529	1.36	0.52–3.55	34	89.47	40	85.11	0.55	1.49	0.40–5.52
	CT	7	21.88	24	27.59		0.74	0.04–1.92	4	10.53	7	14.89		0.67	0.18–2.49
	TT	0	0	0	0				0	0	0	0			
Alleles	C	57	89.06	150	86.21	0.561	1.30	0.53–3.10	72	94.74	87	92.55	0.57	1.45	0.41–5.14
	T	7	10.94	24	13.79		0.77	0.31–1.88	4	5.26	7	7.45		0.69	0.19–2.45
Dominant	CC	25	78.13	63	72.41	0.529	1.36	0.52–3.56	34	89.47	40	85.11	0.55	1.49	0.40–5.52
	CT-TT	7	21.88	24	27.59		0.74	0.04–1.92	4	10.53	7	14.89		0.67	0.18–2.49
Recessive	CC-CT	32	100	87	100				38	100	47	100			
	TT	0	0	0	0				0	0	0	0			
rs1303															
Codominant	TT	14	43.75	39	44.83	1	1		17	44.74	23	48.94	1		
	GT	16	50	38	43.68	0.656	1.17	0.50–2.73	13	34.21	15	31.91	0.928	1.17	0.44–3.10
	GG	2	6.25	10	11.49		0.56	0.11–2.86	8	21.05	9	19.15		1.20	0.38–3.76
Alleles	T	44	68.75	116	66.67	0.761	1.1	0.59–2.04	47	61.84	61	64.89	0.681	0.88	
	G	20	31.25	58	33.33		0.91	0.49–1.68	29	38.16	33	35.11		1.14	0.61–2.14
Dominant	TT	14	43.75	39	44.83	0.916	0.96	0.42–2.17	17	44.74	23	48.94	0.699	0.84	0.36–1.99
	GT-GG	18	56.25	48	55.17		1.04	0.46–2.36	21	55.26	24	51.06		1.18	0.50–2.79
Recessive	TT-GT	30	93.75	77	88.51	0.399	1.95	0.40–9.42	30	78.95	38	80.85	0.827	0.89	0.31–2.58
	GG	2	6.25	10	11.49		0.51	0.11–2.48	8	21.05	9	19.15		1.13	0.39–3.27

FE-S = Patients with COPD related to tobacco smoking frequent exacerbators, NEX-S = Patients with COPD related to smoking non-exacerbators. FE-BB = Patients with COPD related to biomass burning frequent exacerbators. NEX-BB: Patients with COPD related to biomass burning non-exacerbators. GF = Genotype frequency, AF = Allele frequency.

**Table 5 ijms-21-00195-t005:** Characteristics of SNPs.

Gene	SNP	Alleles	PiM Subtypes	MAF in Mexicans (LA)	Chromosome Position	Aminoacid Change	Consequence
*SERPINA1*	rs709932	C/T	M2/M4	0.06	chr14:9438286	R (Arg) 102 H (His)	Missense variant
	rs1303	T/G	M3	0.28	chr14:94378506	E (Glu) 376 D (Asp)	Missense variant

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
