# Peer review of "Haplotype in SERPINA1 (AAT) Is Associated with Reduced Risk for COPD in a Mexican Mestizo Population"

_ijms, 2019, doi:10.3390/ijms21010195_

Round 1

Reviewer 1 Report

The manuscript entitled: “Haplotype in SERPINA1 (AAT) is associated with reduced risk for COPD in a Mexican mestizo population” is an analysis of two SNPs in SERPINA1 gene in relation to COPD caused by tobacco smoke but also what is the most valuable part of this work to COPD associated with biomass-burning exposure. Although the authors did not find the association of these SNP with CODP morbidity of disease exacerbation but they do find the correlation of CT haplotype with reduced risk for smoking related COPD. The manuscript is well written, with adequate statistics however I found some flaws:

The most important difficulty with interpretation of these data are missed values especially for smoking status of biomass-burning related COPD group or what I assume is more difficult to upgrade the data about biomass-burning exposure in smoking associated COPD. The lack of analysis taking into account the influence of these factors is an important issue of this manuscript. The smoking associated COPD group is relatively small compared to others – authors mention this problem in study limitation section. Table 1 caption: where are the HS – healthy subjects? The data are shown as median or mean value? The additional comparison of smoking related vs biomass-burning related groups would be helpful (Table 1 and Table 2).

Author Response

Thank you very much for your kind comments and especially by your conscientious revision.

Regarding your advice on the lack of analysis of biomass-burning exposure in smoking-associated COPD, now we have clarified in the methods section that those patients with COPD related to tobacco smoking have no history of biomass-burning exposure, and those patients with COPD related to biomass-burning exposure are never-smokers. Regarding Table 1, we are sorry for that mistake. Now we have deleted “HS: healthy subjects” from the footnote table and we also clarified that data are shown in median values in Tables 1 and 2. Now we have included an additional comparison between smoking-related COPD biomass-burning related COPD groups in Supplementary Table S1 and S2.

Reviewer 2 Report

I have read the article entitled „Haplotype in SERPINA1 (AAT) is associated with reduced risk for COPD in a Mexican mestizo population” with great interest. I have a few comments:

Please explain COPD-S, COPD-BB, SWOC and BBES in the abstract. Introduction, 1st Please change “important exposure” to “significant exposure”. Please rephrase the last paragraph of Introduction as it does not make sense currently. COPD development is associated with socioeconomic status, childhood asthma, symptoms of chronic bronchitis and frequent infections in childhood. The results should be adjusted for these factors. Discussion must be shortened as several paragraphs are not related to the Results. Was pre- or post-bronchodilator spirometry used at diagnosis? Did any of the biomass-exposed group smoke less than 10 PY? Instead of “lung function decline” use “lung function”. Please, provide power calculations.

Author Response

Thank you very much for your meticulous revision. Now we have made the respective changes:

We have clarified in the abstract that COPD-S, COPD-BB, SWOC, and BBES meanings. We have changed “important exposure” with “significant exposure” in the first paragraph of the introduction. The last paragraph of the introduction section was changed. We agree with your comment about some adjusts, several studies have demonstrated certain participation between socioeconomic status, childhood asthma and frequent respiratory infections in childhood with COPD development. However, firstly, each comparison was done among subjects from the same communities, i. e. COPD-BB and exposed without COPD belong to the same municipalities; regarding tobacco-smoking comparison, cases and controls attend to COPD and smoking cessation institutional clinics, which is a third-level hospital, attending people without social security; reducing the possibility to populational stratification among comparison groups. Finally, our study design was nested in early-COPD detection campaigns ("Diagnóstico oportuno de EPOC / Respirar sin humo" as is stated in the methods section); we didn’t collect these data, this is the reason which we did not adjust our results by those variables. In addition, we do not ask for re-contact, making not possible looking in a prospective way these variables. In the current manuscript version, we have re-formulated the cases-control enrollment description. Now we have shortened the Discussion section. Now we have clarified in the methods section that COPD diagnosis was made using a ratio FEV1/FVC <70% after post-bronchodilator use. Regarding your question: “Did any of the biomass-exposed group smoke less than 10 PY?” this is an important concern, (requested by another reviewer too); now we have clarified in the methods section that we just included patients with COPD related to tobacco smoking who have no history of biomass-burning exposure and patients with COPD related to biomass-burning exposure without a background of active or passive smoking. Now we have changed “lung function decline” by “lung function” in the methods section. Power calculations have been included in the methods sections. Regarding the retrospective power analysis of the 2 associated polymorphisms, according to MAF for each SNP to reach an 80% statistical power (CI 95%, OR=2.0), in tobacco-smoking comparison we need 196 case-patients and 451 controls for the rs709932; while 108 case-patients and 249 controls are needed for rs1303. In this comparison, we have a sample-size higher than required (297 cases and 674 controls).

In the biomass-burning comparison for the rs709932, the statistical power calculation was performed with 96 cases and 288 controls; for this SNP we have a sample-size higher than required (178 cases and 551 controls). Conversely, for the rs1303, according to its MAF, 299 cases and 897 controls are required to reach a statistical power of 80%. In this last SNP, we have a sample-size lower than required. Now, this has been pointed out as a limitation in the study, which must be taken into consideration and possibly require later populational validation.

Reviewer 3 Report

This manuscript aims to defines the influence of SNPs in the SERPINA1 gene associated on COPD in a Mexican mestizo population.  The authors utilized a relatively large patient population and performed several statistical stratifications/analyses.  The manuscript has potential to add new information to the field.  Below are comments that can help strengthen and improve the manuscript.

Abstract:

Please indicate what PiS and PiZ variants are.  For example, please indicate which gene this is.

Please defined the groups prior to abbreviating, for example, what does S, BB, SWOC, and BBES indicate in the following sentence “…COPD-S (297), COPD-BB (178), SWOC (674) and BBES (551)”.

Introduction:

The following sentence would be better separated into two sentences “In accordance with Global Initiative for Chronic Obstructive Lung Disease (GOLD), Chronic Obstructive Pulmonary Disease (COPD) is a common, preventable and treatable disease characterized by persistent respiratory symptoms and airflow limitation, caused by an important exposure to noxious particles or gases, being the most important tobacco smoking and biomass-burning exposure [1].”

Please provide a reference for the following sentences “Nowadays COPD is considered the fourth leading cause of death worldwide.”

Please list a reference for the following sentence “Alpha-1 antitrypsin deficiency (AATd) is the most important genetic risk factor associated with COPD”.

Please refine this sentence as it is not clear.  For example, variants in SERPINA referred to as…. “Despite these variants represents approximately 95% of the AATd cases, there are several rare other variants such as Siiyama, Mmalton, Mprocida, Mheerlen, Mmineral springs, Mnichinan, Pduarte, Wbethesda Zaugsberg, and Zbristol that could confer risk to COPD, but there are not enough studies to support it [4].

This sentence is not clear and should be refined. For example, perhaps eliminate the ‘suggesting that exists’.

“Recently, Fernández-Acquier and colleagues [7] described that after genetic sequencing found discordance between genotype and serum AAT levels; suggesting that exists an important proportion of COPD patients with different genetic variants different to PiS and PiZ.”

Results

Table 1, the proportion of individuals that were male in the COPD-BB and BBES group appear to be extraordinarily low.  Please confirm these values are correct. 

Figure 1.  Please provide a more detailed figure legend in order to provide a better understanding of Figure 1.

Discussion

Please provide a few sentences/paragraph regarding the striking sex differences observed among demographics. The authors should clarify whether this is due to a biological difference or a limitation of experimental design (for example patient enrollment practices).

Materials and Methods

Please provide enrollment criteria for the study as well as the years that the study encompassed.

Similarly, please clearly list/provide and exclusion criteria.

Author Response

Thank you very much for your conscientious revision. Now we have made the changes that were required.

Abstract:

Now we have indicated in the abstract that PiS and PiZ variants belong to the SERPINA1 We have clarified in the abstract COPD-S, COPD-BB, SWOC, and BBES meanings.

Introduction:

The first paragraph was divided into two separate sentences. Now we have provided a reference for the following sentence “Nowadays COPD is considered the fourth leading cause of death worldwide” in the Introduction section. Now we have provided a reference for the following sentence “Alpha-1 antitrypsin deficiency (AATd) is the most important genetic risk factor associated with COPD” in the Introduction section. We have clarified the sentence “Despite these variants represents approximately 95% of the AATd cases…” in the Introduction section. We have clarified the sentence “Recently, Fernández-Acquier and colleagues…” in the introduction section.

Results

Regarding the proportion of individuals that were male in the COPD-BB and BBES group appear to be extraordinarily low, the values are correct. Now we have an appropriate paragraph in the discussion section explaining this phenomenon. According to your request, a more detailed Figure 1 was depicted.

Discussion

Now we have an appropriate paragraph in the discussion section explaining this phenomenon.

Material and Methods

Now we have described the periods in which patients and exposed subjects (for both comparisons) were included. In addition, a flowchart including exclusion criteria is provided as a supplementary figure.

Round 2

Reviewer 2 Report

I am happy with the changes and suggest acceptance.